# The proteomic landscape of stool-derived extracellular vesicles in patients with pre-cancerous lesions and colorectal cancer
Emmalee J. Northrop-Albrecht [1] ✉, Yohan Kim[2], William R. Taylor[1], Shounak Majumder[1], John B. Kisiel[1] & Fabrice Lucien [2,3] ✉

Colorectal cancer (CRC) is the 2nd most fatal cancer in the United States, but when detected early it is highly curable. Stool-derived extracellular vesicles (EVs) are a novel biomarker source that could augment the sensitivity for detection of CRC precursors. However, standardization of isolation methods for stool-derived EVs remains underexplored. We previously reported that size-exclusion chromatography (SEC) followed by ultrafiltration (UF-100kDa) was suitable for human stool supernatant EV isolation. In this study, we first assess alternative EV concentration methods (ultrafiltration [UF]; 10 kDa, 30 kDa, 50 kDa, 100 kDa and speed vacuum [SV]). Second, we investigate the host/bacterial EV proteomes by mass spectrometry. We report no difference in recovery, RNA and soluble protein contamination among concentration methods. Proteomic analysis reveals a diverse bacterial proteome, while human-derived proteins are more abundant. Specifically, pancreatic enzymes are among the most abundant proteins, further exploration revealed that zymogen granules are likely co-isolated in stool EV preparations. To enable discovery of EV-based molecular signatures of CRC precursors with high sensitivity, immunocapture strategies will likely be needed. Notably, we identified 10 surface proteins that may serve as candidates for the purification of colon-derived EVs. This work serves as framework for the future discovery and validation of EV-based biomarkers for CRC.

Colorectal cancer (CRC) remains the 2nd most fatal cancer in the United States, despite advancements in screening and therapies[1]. The 5-year survival rate for CRC when detected early (stages 1 and 2) is around 90%, so early diagnosis is critical[1]. Adenomatous polyps or sessile serrated lesions can eventually develop into cancer in a span of 10–15 years. The multitarget stool DNA (mt-sDNA) test is a non-invasive screening option that has high detection rates for CRC, but only 43% sensitivity at detecting advanced precancerous lesions[2]. Extracellular vesicles (EVs) may be a potential source of complementary biomarkers that could enhance the detection of advanced precancerous lesions, which would be anticipated to improve CRC prevention.

Extracellular vesicles are small particles (~30 nm to 10 μm) that are secreted by all living cells into the extracellular milieu. They are enclosed by a phospholipid bilayer and contain nucleic acids, proteins, lipids, and metabolites that serve an important role in cell-to-cell communication[3]. There are several techniques that can separate EVs from soluble components, these include: differential centrifugation, density gradient centrifugation, ultrafiltration (UF), size-exclusion chromatography (SEC), immunocapture, and polymer-based precipitation[4]. However, there is no gold standard for EV isolation as each method is variable in purity and recovery among different biofluids.

While EVs have been isolated from numerous biological matrices, less is known about their isolation and characterization from human stool samples. It is a preferred biospecimen for the discovery of early CRC precursors which exfoliate cells and cellular constituents into the fecal stream. We previously conducted a thorough head-to-head comparison of EV isolation/enrichment methods from human stool specimens using the following criteria: particle recovery, reproducibility, purity, protein yield/expression, and RNA composition[5]. We concluded that SEC was the most suitable EV isolation method for stool supernatant based on the parameters above. SEC results in EVs being distributed across multiple fractions. Most downstream applications are not conducive with large sample volumes; therefore, the EV rich fractions are often pooled and concentrated using an additional isolation technique such as ultrafiltration or ultracentrifugation[6–9]. For the previous

[1]Division of Gastroenterology and Hepatology, Mayo Clinic, Rochester, MN, USA. [2]Department of Urology, Mayo Clinic, Rochester, MN, USA. [3]Department of Immunology, Mayo Clinic, Rochester, MN, USA. ✉e-mail: Northrop.emmalee@mayo.edu; Lucien-Matteoni.Fabrice@mayo.edu

study, we selected 100 kDa ultrafiltration centrifugal filters following SEC because it would likely result in the highest purity. The influence of ultrafiltration molecular weight cut-off (MWCO) on EV recovery and purity has been previously investigated for blood and urine, but not for stool supernatant[10].

Stool is comprised of water, protein, undigested fats, polysaccharides, ash, undigested food residues, and a variety of bacteria[11]. Since EVs are released by all three domains of life (eukaryotes, bacteria, and archaea), stool EV proteomes are complex and dynamic[12]. The EV proteomic landscape in human stool supernatant has been underexplored. Of the studies that have been conducted in this area of research, they differ in the type of disease studied, EV isolation method and protein detection application used. The fecal bacterial EV proteome has been previously explored in several cancers, none of which were colorectal[13]. Wu and others (2019) carried out mass spectrometry on stool-derived EVs in a limited number of Crohn's patients using differential centrifugation and filtration[14]. Zhang and others (2023) applied previous literature and bioinformatic strategies to select and test human fecal EV CRC biomarker candidates using differential centrifugation[15]. Overall, this work, is the first to examine the proteomic landscape in stool EV preparations among cancer-free, pre-cancerous lesions, and colorectal cancer patients using our optimal EV isolation technique and mass spectrometry.

In the current study, we determined that EV concentration methods post-SEC had limited impact on stool EV derived protein and RNA recovery. Our comprehensive analysis of the bacterial and host proteome of stool EV preparations from cancer-free individuals and patients with different stages of CRC identified tissue markers of the GI tract including colon-associated protein signatures that could be further investigated as biomarkers to enhance the sensitivity for detection of pre-cancerous lesions. Additionally, we uncovered that zymogen granules containing highly abundant enzymes are likely co-isolated in stool EV preparations.

## Result
### Concentration methods had limited impact on stool EV protein and RNA characteristics

By using transmission electron microscopy, we confirmed the presence of extracellular vesicles after isolation and examined purity. Similar to our previous work, we did not observe lipoprotein particles in EV samples[5]. As MWCO increases, presence of visual contaminants appears to lessen (Fig. 1A, Supplementary Fig. 1). There is little if any literature on using a speed vacuum as an alternative EV concentration method. Speed vacuum EV preparations contained intact EVs but slightly more visual contaminants than UF-100kDa (Supplementary Fig. 1).

Nanoscale flow cytometry revealed no difference in SEC recovery rates among Izon's 35 nm and 70 nm SEC columns (Mean ± SD): 35 nm: 56.92% ± 10.02, 70 nm: 59.38% ± 5.99; $P > 0.51$; (Supplementary Fig. 2A). Additionally, UF recovery was significantly improved with 15 ml centrifugal filters compared to 4 ml centrifugal filters (Mean ± SD): 4 ml: 36.41% ± 18.62, 15 ml: 58.10% ± 21.90; $P < 0.002$; (Supplementary Fig. 2B). Therefore, 70 nm SEC columns and UF-15ml centrifugal filters were used for subsequent downstream analyses. MRPS was used to measure particle size distribution and recovery rates among three patient samples for all concentration methods. The SEC recovery rates were (Mean ± SD): 46.45% ± 18.29 (sample 1), 63.16% ± 11.23 (sample 3), and 59.36% ± 18.57 (sample 4) (Supplementary Fig. 2C). The average post-SEC recovery rates among all patients were (Mean ± SD): 10 kDa: 62.35% ± 16.71, 30 kDa: 64.76% ± 21.27, 50 kDa: 67.67% ± 16.32, 100 kDa: 39.39% ± 9.44, SV: 68.06% ± 7.17 (Supplementary Fig. 2D, E). Speed vacuum had improved post-SEC recovery compared to UF-100kDa. Total particle recovery (SEC + UF/SV) ranged from 15% to 65%. The average total recovery rates among all patients were (Mean ± SD): 10 kDa: 34.56% ± 14.34, 30 kDa: 32.66% ± 11.14, 50 kDa: 35.71% ± 14.94, 100 kDa: 26.61% ± 4.87, SV: 38.34% ± 25.19 (Fig. 1B, Supplementary Fig. 2F). Fig. 1C revealed that regardless of concentration method, ~80% of the particles measured were between 75 nm and 105 nm.

Co-isolation of non-vesicular RNA is a factor affecting downstream RNA analysis[16]. Additionally, non-vesicular RNA can bind to ribonucleoprotein

complexes (RNPs) and lipoproteins[17]. For these reasons, we treated EV fractions from each of the concentration methods with Proteinase K and RNase A in order to disrupt RNPs and digest any RNA outside the vesicles (Supplementary Fig. 3A). Vesicular RNA post-enzymatic treatment ranged from 8% to 43% indicating that most of the RNA from EV samples are non-vesicular in origin. The average percent vesicular RNA among all patients were (Mean ± SD): 10 kDa: 19.37% ± 12.00, 30 kDa: 19.53% ± 10.86, 50 kDa: 18.70% ± 8.56, 100 kDa: 23.29% ± 13.66, SV: 16.05% ± 5.18 (Fig. 1D, Supplementary Fig. 3B). The average vesicular RNA yield among all patients were (Mean ± SD): 10 kDa: 29.15 ng ± 5.79, 30 kDa: 27.20 ng ± 3.98, 50 kDa: 24.16 ng ± 6.16, 100 kDa: 29.65 ng ± 4.79, SV: 28.12 ng ± 6.07, (Supplementary Fig. 3C, D). Speed vacuum had increased RNA yield compared to UF-50kDa. Supplementary Fig. 3E shows an example of the fragment size distribution profile pre- and post-enzymatic treatment on a bioanalyzer. A similar profile was observed across concentration methods with most of the fragments being between 25 and 175 nucleotides.

Protein yield was negatively correlated with the MWCO of ultrafiltration columns. The average total protein yield across samples for 100 µl concentrate were (mean ± SD): 55.26 µg ± 10.48 (SV), 37.17 µg ± 16.60 (10 kDa), 29.28 µg ± 3.72 (30 kDa), 22.45 µg ± 3.76 (50 kDa), and 17.16 µg ± 4.53 (100 kDa) (Fig. 1E, Supplementary Fig. 4A). The level of non-EV protein contamination was estimated by calculating the ratio of particle count and protein yield[18]. Speed vacuum had a lower ratio compared to UF-30kDa and UF-50kDa, indicating more soluble protein contamination (Fig. 1F, Supplementary Fig. 4B).

### The human proteome landscape of stool EV preparations
We used mass spectrometry to examine the impact of EV concentration methods on the protein composition of stool EV preparations obtained from two healthy individuals (Supplementary Data 1). We included soluble protein-rich fractions (F18-F20) to evaluate the co-isolation of soluble proteins. We also performed mass spectrometry on a larger cohort of stool EV samples ($n = 63$ patients) to gain a better understanding of the composition of the stool EV proteome (Supplementary Data 2).

Total protein abundances for each sample/concentration method are depicted in Fig. 2A and Supplementary Fig. 5A. For sample 3, the following number of human proteins were detected: 149 (10 kDa), 132 (30 kDa), 154 (50 kDa), 136 (100 kDa), 79 (SV), 110 (SF), and 208 (Crude SS) (Fig. 2B, Supplementary Fig. 5B). For sample 4, the following number of human proteins were detected: 78 (10 kDa), 75 (30 kDa), 75 (50 kDa), 74 (100 kDa), 39 (SV), and 106 (SF) (Fig. 2B, Supplementary Fig. 5B). Overall, for samples 3 and 4, SV had 41.9% and 47.3% less proteins detected when compared to their 100 kDa sample. For the concentration method comparison, there were 21 previously established human EV markers detected in EV samples[19-21] (Supplementary Data 1). While in the crude stool supernatant sample there were 14 of the EV markers detected. There was no difference in EV iBAQ abundance and % EV marker abundance among concentration methods within the same patient or across patients (Supplementary Fig. 5C–F). While not significant, an enrichment in EV markers was observed in EV preparations compared to soluble fractions (Supplementary Fig. 5F). Each sample had detectable protein levels for at least three EV markers with all EV preparations positive for LGALS3BP, RPS27A, and JCHAIN. Additionally, the EV markers CD63, TSG101, LGALS3BP were detected by western blot in all concentration methods (Supplementary Fig. 5G). There were 32 EV markers detected in the larger cohort, with the most common being JCHAIN, LGALS3BP, GAPDH, and A2M.

There was a total of 363 unique human proteins detected across all samples in the larger cohort, with an average of 88 ± 27 human proteins detected among each sample (mean ± SD; Fig. 2B). Three proteins (CELA3A, IGKC, CTRC) cover 50% of the total protein abundance and only 15 proteins had a relative abundance above 1% (Fig. 2C, D). Approximately two thirds of the human-derived proteins were only detected in 25% or less of the samples, thus indicating the heterogenous nature of stool EV preparations (Fig. 2E). There were 40 proteins that were present in >75% of the stool EV samples (Fig. 2F). For the enrichment

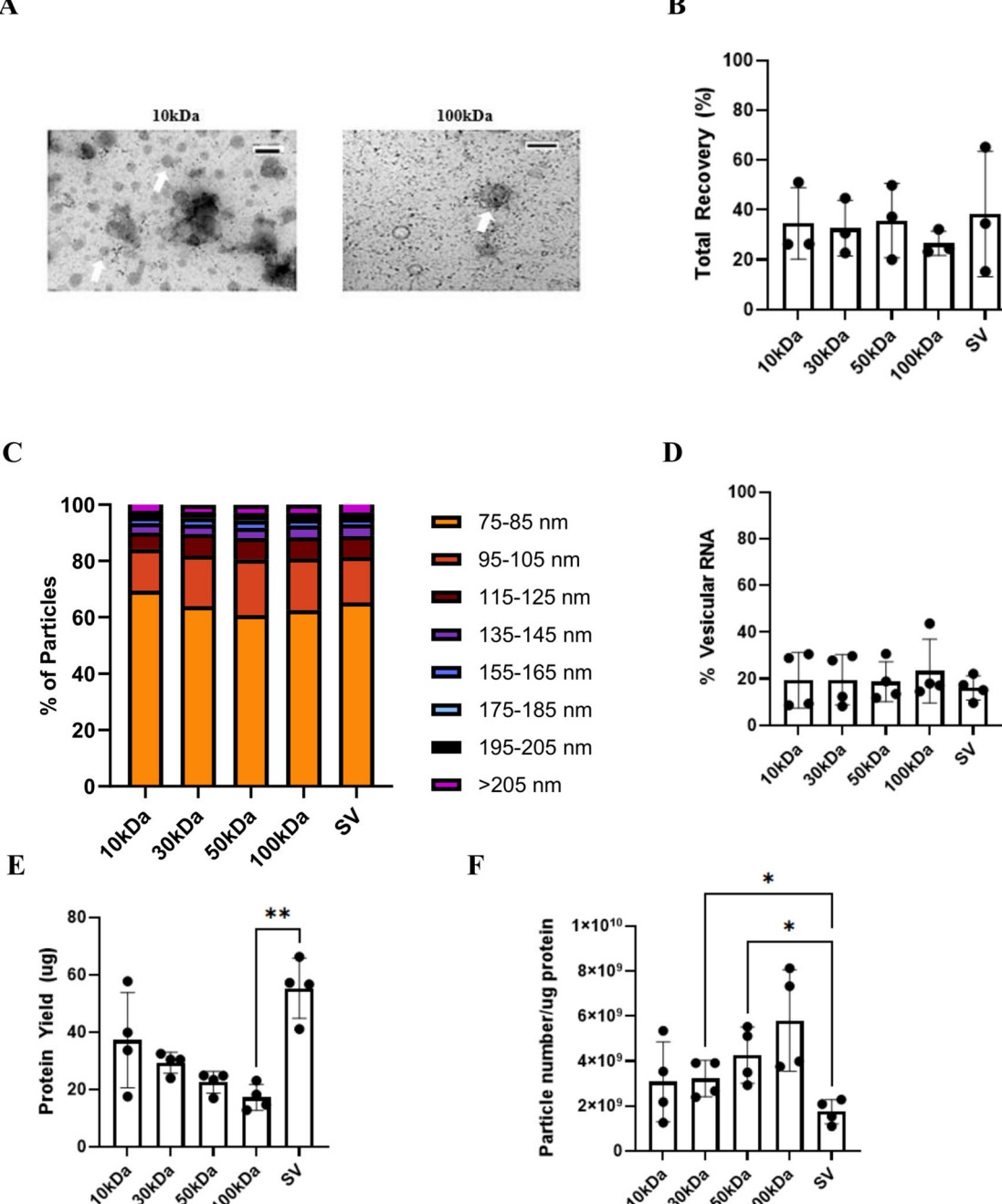

**Fig. 1 | Concentration methods have limited impact on stool EV derived protein and RNA recovery. A** Representative transmission electron microscopy images showing EVs that were isolated by SEC and further concentrated by ultrafiltration (10 kDa and 100 kDa) at 80kx (scale bar: 200 nm), with white arrows indicating individual EVs. **B** Total particle recovery as measured by MRPS for the different concentration methods, Mean ± SD, Friedman Test. **C** Bar chart showing the percent of particles that fall within a given particle size range. **D** Percent vesicular RNA among concentration methods, Mean ± SD, Friedman Test. **E** Total protein yield (µg) across the different EV concentration methods measured by BCA assay, Mean ± SD, Friedman Test, **$p < 0.01$. **F** Ratio of total particle number and total protein yield, a higher ratio suggests less co-isolated soluble protein contamination, Mean ± SD, One Way ANOVA, *$p < 0.05$.

pathway analysis, some of the top GO biological processes associated with these proteins were related to digestion, neutrophil aggregation/chemotaxis, and defense response (Fig. 2G). Top KEGG pathways were pancreatic secretion and protein digestion/absorption (Fig. 2G). The majority of the detected proteins were intracellular (67%) followed by secreted (21%) and plasma membrane-bound (12%; Fig. 2H). We also classified proteins by class and function to compare the protein composition of stool EVs with two commonly studied biofluids, plasma and urine (Fig. 2I). Stool EV preparations have a distinct proteome landscape, primarily composed of immunoglobulins (42%) and hydrolases (41%).

To further evaluate sample purity, we compared EV preparations with soluble protein-rich fractions in order to identify soluble proteins that are lost or co-isolated during SEC and UF (Supplementary Fig. 6A, B). Interestingly, few proteins were unique to EV fractions which include basic proline-rich salivary gland proteins (PRB1, PRB3, PRB4, PRH1, SMR3B), these proteins originate from minor salivary glands and play a role in food digestion[22]. A large proportion of proteins were shared between EV preparations (UF-100kDa) and soluble protein fractions (sample 3: 64 proteins/35%, sample 4: 54 proteins/43%); however, the abundance of these overlapping proteins was reduced in EV fractions compared to SF fractions by

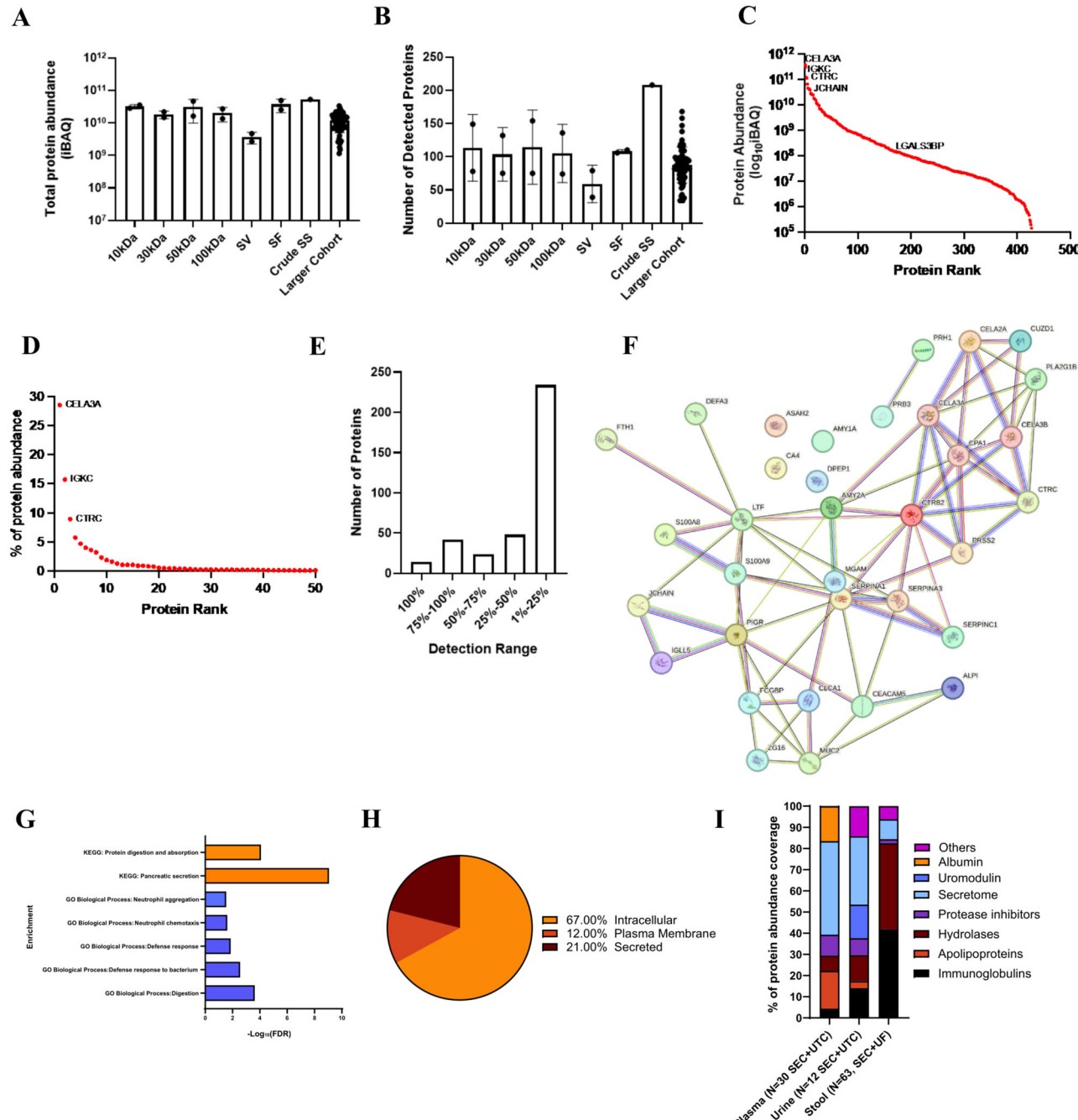

**Fig. 2 | Human proteomic landscape of stool-derived extracellular vesicles.**
**A** Total protein abundance for samples submitted for mass spectrometry (iBAQ values), Mean ± SD. **B** Number of human proteins detected for each sample, Mean ± SD. **C** Protein abundance (iBAQ) for each protein detected in larger cohort oriented by rank. **D** Percent of total protein abundance per protein detected in larger cohort oriented by rank. **E** Detection range for human proteins in the larger cohort of 63 stool EV samples. **F** String diagram depicting interactions of proteins identified in 75% or more of samples. **G** Enriched GO and KEGG processes among human proteins detected in 75% or more of samples in larger cohort. **H** Subcellular location for human proteins identified in the large cohort of 63 stool EV preparations. **I** Protein coverage among stool, urine, and plasma EV preparations.

72% and 80% in sample 3 and 4 respectively (Supplementary Data 1). Secreted protease inhibitors from the SERPIN family (SERPINB1, SERPINB6, SERPINA3, SERPINA1, SERPINC1) and the liver-derived soluble protein Transthyretin (TTR) were among the most abundant proteins in soluble protein fractions, while their abundance in EV preparations were greatly reduced (Supplementary Fig. 6C). These proteins could potentially serve as protein markers of purity in stool EV isolation. Surprisingly, the secreted pancreatic hydrolases CTRC and CELA3A were among the most abundant proteins in EV fractions (Supplementary Fig. 6D, E). These proteins appear to be enriched during the EV isolation process given their

abundance is greatly reduced in soluble protein-rich fractions (Sample 4: 49.3% vs 7.9%, Sample 5: 81.0% vs 43.3%; Supplementary Fig. 6E). In the larger cohort of patients, the mean relative abundance of CTRC and CELA3A combined was 39.0% (range: 3.8%–71.6%; Supplementary Fig. 6E).

**Highly abundant pancreas-derived zymogen granules are co-isolated in stool EV preparations**
Our data revealed that hydrolases are highly abundant in stool EV samples. The number of hydrolases detected in each sample is depicted in Fig. 3A and

Supplementary Fig. 7A. For the larger cohort, more than half of the hydrolases are only detected in less than 25% of samples, with four (CTRC, CELA3A, CPA1, PRSS2) being present in all samples (Fig. 3B). Furthermore, of the 88 hydrolases detected in stool samples, 15 are secreted by the gastrointestinal tract. These GI secreted hydrolases made up anywhere from 4%–80% of the total protein abundance (mean: 44%) within each sample of the larger cohort (Fig. 3C, Supplementary Fig. 7B). We further characterized hydrolases by tissue specificity, and the pancreas had the most specific/ enriched enzymes (Fig. 3D).

Zymogen granules are storage organelles in the exocrine pancreas that allow the sorting, packaging and regulated apical secretion of digestive enzymes[23]. Several of the proteins identified in stool EV preparations have been associated with zymogen granules these include: ZG16, CUZD1, CLCA1, GP2, DMBT1, ZG16B, CTRC, CTRB1, PLA2G1B, PNLIP, SMPDL3A, DPEP1, CPA2, CPA1, AMY1A, PRSS2, CLPS, CELA2A, CELA3B, and CELA3A[23–25]. Notably, three proteins (GP2, CUZD1, DPEP1) are membrane-bound, thus providing further evidence of the co-isolation of secretory zymogen granules during the EV isolation process in stool. Using western blot, we initially tested stool EV preparations for the presence of CELA3A and GP2, a major zymogen granule membrane protein alongside pancreatic juice derived EVs as a positive control[26–30] (Fig. 3E). We then performed dot blot analysis with intact stool-derived EVs to further evaluate the subcellular location of these proteins. For CELA3A, protein abundance increased when exposed to Tween-20 detergent as demonstrated by the darker dot (Fig. 3E). This suggesting that CELA3A is found mainly within the vesicles. For GP2, no visible difference in abundance among tween and non-tween treated samples was observed (Fig. 3E). Flow cytometry confirmed that GP2 is expressed on the surface of vesicles and can be detected in stool and pancreatic juice at the single vesicle level (Supplementary Fig. 8A, B). Additionally, PBS only, AB only, and biofluid/biospecimen (pancreatic juice or stool) only samples were run as controls for the experiment (Supplementary Fig. 8C). The concentration of GP2 positive vesicles varied greatly among crude stool supernatant samples (Fig. 3F), and many of them were larger in size compared to pancreatic juice samples. However, GP2 positive vesicles make up a greater percent of the total vesicle abundance in pancreatic juice (6.86%) compared to stool (2.24%; Fig. 3G).

The high concentration of pancreatic serine proteases in stool EV samples reflects endogenous proteolysis which is essential for gastrointestinal physiology. In proteomic studies, it is standard to analyze peptides generated from protein digestion with exogenous trypsin. Endogenous serine proteases have different amino acid cleavage sites which may be missed during peptide analysis. Previous metaproteomics work used a semi-tryptic peptide mining approach for capturing signatures of gut microbial proteolysis[31]. We performed a semi-tryptic analysis on the smaller concentration method cohort in order to determine if the low protein coverage in EV samples is potentially due to the missed detection of peptides cleaved by these highly abundant pancreatic enzymes. Total intensity tended to be increased in semi-tryptic compared to tryptic dataset within each sample ($P = 0.07$; Supplementary Fig. 9A). Total peptide counts from the semi-tryptic analysis were increased when compared to the tryptic dataset (Mean ± SD; Tryptic: 808.4 ± 295.9, Semi-Tryptic: 1527 ± 758.4; $P = 0.01$; Supplementary Fig. 9B). When a minimum two peptide cutoff was applied to EV samples, there was a total of 221 and 215 human-derived proteins detected in the tryptic and semi-tryptic datasets respectively (Supplementary Data 3). There were 51 proteins unique to the semi-tryptic dataset, many of which were immunoglobulin-related (Supplementary Fig. 9C). The comparative analysis of tryptic and semi-tryptic peptides demonstrate that the high concentration of pancreatic proteases does not affect mass spectrometry analysis of EV-associated proteins.

## Colon- enriched and specific proteins are detectable in the stool-derived EV proteome

Given that stool is in direct contact with the gastrointestinal tract, we sought to identify the tissue origin of EVs isolated from stool supernatant. For the concentration method comparison samples, we focused on samples isolated

by SEC and UF-10kDa because it had the most overlapping proteins (73) among the patients. To determine their tissue of origin, we annotated these proteins to the human body proteome map[32]. Out of the proteins that were designated as tissue-enriched and specific, there were 12 associated with the colon, 11 to the small intestine, 15 to the pancreas and 5 to the esophagus (Supplementary Data 1). Of the colon-enriched proteins, 10 of them were localized to the cell surface based on our in-house surfaceome database[33] (Supplementary Data 1). The relative abundance of colon-enriched/specific proteins among concentration methods within each patient are depicted in Supplementary Fig. 10. EV preparations that were concentrated by ultrafiltration were enriched for colon-associated proteins compared to SF (Fig. 4A). Specifically, the abundance of PIGR, ITLN1, and CEACAM5 was increased in UF-10kDa samples compared to SF samples. Additionally, there were three colon-enriched proteins (ZG16, LGALS4, SLC26A3) that were exclusively present in EV samples compared to SF samples. Three colon-enriched surface proteins (SLC26A3, LGALS4, CEACAM5) were further validated using western blot in additional patient samples, one of which is shown in Fig. 4B.

For the larger cohort, there were 15 colon-enriched proteins detected (Table 1, Supplementary Data 2). Colon-associated proteins made up an average of ~6% of the total protein abundance (Fig. 4A). The majority of colon-enriched proteins were plasma membrane-bound (67%) followed by intracellular (27%) and secreted (6%; Fig. 4C). Functional interaction networks confirmed the close relationship among the colon-associated proteins present in the dataset (Fig. 4D). There were seven proteins that were detected in less than 25% of samples, while four proteins were detected in 95% of samples, these included: MUC2, ZG16, PIGR, and CEACAM5 (Fig. 4E). Specifically, CEACAM5 a surface-bound protein, was detected on the surface of stool vesicles through the use of flow cytometry (Supplementary Fig. 11A). Additionally, PBS only, AB only, and stool only samples were run as controls for the experiment (Supplementary Fig. 11B). The concentration of CEACAM5-positive EVs in the larger cohort of stool supernatant samples by nanoscale flow cytometry is depicted in Fig. 4F. CEACAM5-positive EVs made up an average of 2.3% of the total EV population (Fig. 4G).

## Stool EV preparations have a more diverse bacterial proteome, but human-derived proteins are more abundant

For the concentration method comparison cohort, there was a total of 694 bacterial proteins detected in EV preparations, 349 of which were species specific. When SF was included, there were 990 bacterial proteins detected, 494 of which were species specific (Supplementary Data 4). Similar to the human proteome data, SV was the concentration method with the least detected proteins. For sample 3, the following number of bacterial specific proteins were detected with the number of species they annotated to: 87, 17 (10 kDa), 49, 14 (30 kDa), 94, 16 (50 kDa), 58, 14 (100 kDa), 14, 7 (SV), and 35, 13 (SF). For sample 3 UF-10kDa sample, two species (*Faecalibacterium prausnitzii, Alistipes putredinis*) made up ~94% of the total protein abundance, with *Faecalibacterium prausnitzii* being the dominant species (Fig. 5A). For sample 4, the following number of bacterial specific proteins were detected with the number of species they annotated to: 224, 21 (10 kDa), 169, 21 (30 kDa), 178, 21 (50 kDa), 134, 18 (100 kDa), 30, 12 (SV), and 182, 23 (SF). For sample 4 UF-10kDa sample, seven species made up ~94% of total protein abundance, with *Prevotella copri, Faecalibacterium prausnitzii, Ruminococcus bromii* being the most abundant species (Fig. 5B).

For the concentration method comparison cohort, there were 345 redundant bacterial proteins detected in EV preparations. When SF samples were included, there were 496 redundant bacterial proteins from 62 bacterial species detected. For sample 3, there were the following number of proteins detected for each concentration method: 159 (10 kDa), 112 (30 kDa), 128 (50 kDa), 130 (100 kDa), 26 (SV), and 36 (SF). There were 90 of these proteins that were detected in all UF concentration methods regardless of MWCO. For sample 4, there were the following number of proteins detected for each concentration method: 161(10 kDa), 124 (30 kDa), 144 (50 kDa), 115 (100 kDa), 34 (SV), and 163 (SF). There were 99 of these proteins that were detected in all UF concentration methods regardless of MWCO. Some

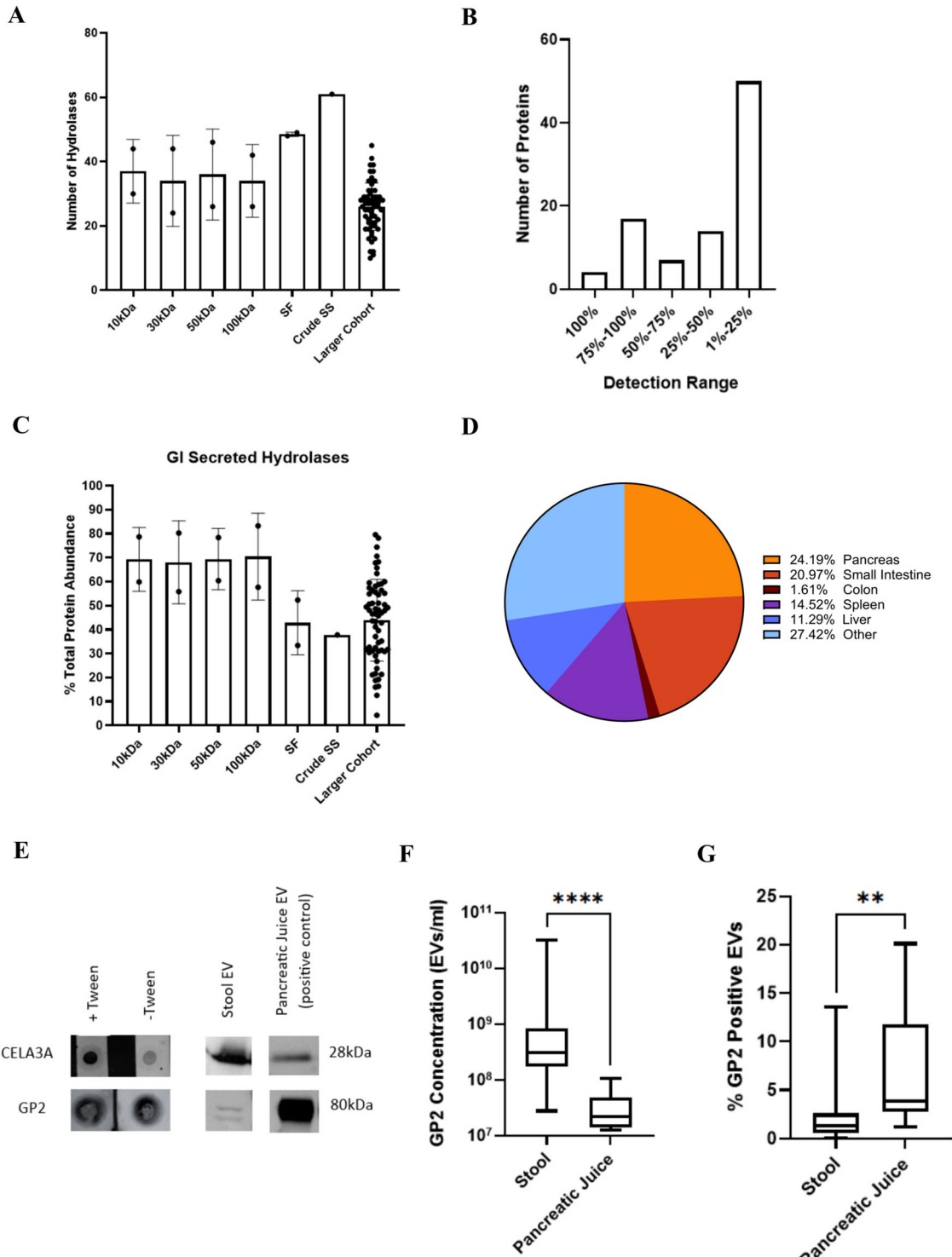

**Fig. 3 | Co-isolation of zymogen granules containing pancreatic enzymes in stool EV preparations. A** Number of proteins with hydrolase activity detected among the different concentration methods and larger cohort, Mean ± SD. **B** Detection range for hydrolases in the larger cohort of 63 stool EV samples. **C** Percent of total protein abundance associated with GI secreted hydrolases among the different concentration methods and larger cohort, Mean ± SD. **D** Pie chart indicating tissue specificity/enrichment of hydrolases detected in larger cohort. **E** Dot blot showing stool-derived EV sample blocked either with or without Tween-20 and incubated with pancreas associated primary antibodies. Additionally western blot was performed using the same antibodies with pancreatic juice derived EVs being used as a positive control. **F** GP2 concentration in stool and pancreatic juice measured by flow cytometry, Mann–Whitney Test, ****$p < 0.0001$, the boxes and the whiskers in this plot and hereafter indicate the first and third quartiles, the medians, and the minimum and maximum values, respectively. **G** Percent GP2 positive EVs in stool and pancreatic juice measured by flow cytometry, Mann–Whitney Test, **$p < 0.01$.

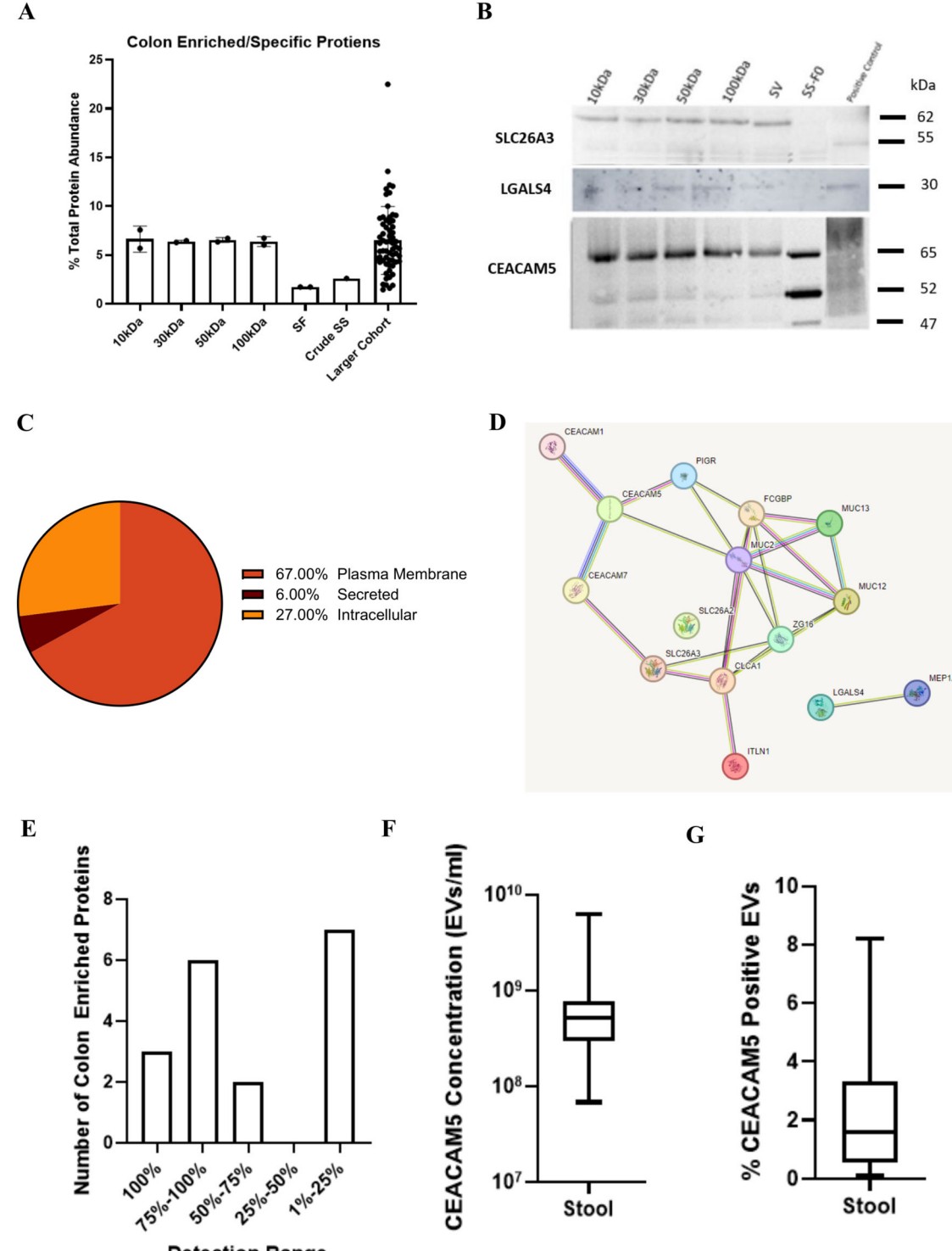

**Fig. 4 | Colon-enriched/specific proteins identified in stool EV preparations.**
**A** Percent of total protein abundance associated with colon enriched/specific proteins among the different concentration methods and larger cohort, Mean ± SD.
**B** Western blot (15ug) showing colon enriched protein expression among the five concentration methods and pre-isolation sample (stool supernatant). The positive control is from a sample that was submitted for mass spectrometry. **C** Subcellular location for colon enriched/specific proteins identified in the larger cohort of 63 stool EV samples. **D** String diagram depicting interactions of proteins specific/enriched in the colon. **E** Detection range for colon enriched/specific proteins in the larger cohort of 63 stool EV samples. **F** CEACAM5 concentration in stool measured by flow cytometry. **G** Percent CEACAM5 positive EVs in stool measured by flow cytometry.

common proteins that overlapped among the two patients included: OmpA family protein, TonB-linked outer membrane protein, and SusD family protein. We selected ompA and flagellin as bacterial EV markers to further validate by western blot in additional patients, one of which is shown in Fig. 5C. The ompA protein was previously recognized as a bacterial-derived EV protein marker[34]. Flagellin has been identified in gram-negative bacterial EVs as a major subunit of flagella and a potential contaminant in stool EV preparations[35,36].

While the bacterial proteome is more diverse and complex, human-derived proteins are found more abundantly in stool (Fig. 5D). The median

**Table 1 | Colon enriched/specific proteins that were detected in stool EV preparations by mass spectrometry**

| Protein Name | Tissue Enrichment/Specific | Abbreviation | Surfaceome |
|---|---|---|---|
| Carcinoembryonic antigen-related cell adhesion molecule 1 | Colon Transverse | CEACAM1 | X |
| Carcinoembryonic antigen-related cell adhesion molecule 5 | Esophagus, Colon Transverse | CEACAM5 | X |
| Carcinoembryonic antigen-related cell adhesion molecule 7 | Colon Transverse | CEACAM7 | X |
| Calcium-activated chloride channel regulator 1 | Small intestine, Colon Transverse | CLCA1 | |
| IgGFc-binding protein | Colon Transverse, Small intestine | FCGBP | |
| Intelectin-1 | Colon Transverse | ITLN1 | |
| Galectin-4 | Colon Transverse, Small intestine | LGALS4 | X |
| Mucin-2 | Small intestine, Colon Transverse | MUC2 | |
| Polymeric immunoglobulin receptor;Secretory component | Colon Transverse, Small intestine | PIGR | X |
| Chloride anion exchanger | Colon Transverse | SLC26A3 | X |
| Zymogen granule membrane protein 16 | Colon Transverse, Small intestine | ZG16 | |
| Meprin A subunit alpha | Colon Transverse, Small intestine | MEP1A | X |
| Mucin-13 | Colon Transverse, Small intestine | MUC13 | X |
| Solute carrier family 26 member 2 | Colon Transverse, Small intestine | SLC26A2 | X |
| Mucin-12 | Colon Transverse | MUC12 | X |

total abundance of all concentration methods for each sample was calculated for both human and bacterial proteins, and the median fold difference for samples 3 and 4 were 6.86 and 2.46. When the top four highly abundant secreted proteins were removed, human total protein abundances and median fold differences drop significantly to 2.36 and 0.25 (Fig. 5E). Besides human and bacterial- derived proteins, we detected proteins from food consumption and derived from chicken, cow, pig, potato, radish, almond, orange to name a few.

## Discussion

Few studies have investigated the potential of stool-derived EVs in CRC diagnosis, but a comprehensive analysis of the stool EV proteome is still lacking[15,37]. Importantly, the pre-analytical variables for isolation of stool-derived EVs have not been well studied and standardized. Our previous work evaluated different EV isolation strategies, and the combination of size-exclusion chromatography and ultrafiltration (UF-100kDa) was the optimal separation method based on purity and recovery parameters[5]. In this study, we evaluated the impact of ultrafiltration MWCO post-SEC on recovery, purity, RNA composition, protein yield, and soluble protein contamination in a small subset of stool samples. Then, using the optimal methodology, we performed mass spectrometry-based proteomics to uncover the human and bacterial proteome of stool-derived EVs isolated from a larger cohort of 63 patients.

Four UF-MWCOs (10 kDa, 30 kDa, 50 kDa, 100 kDa) were evaluated, and ~60% of particles were recovered from all MWCOs except UF-100kDa which was around 40%. Other groups reported that UF-MWCO can influence particle recovery when EVs were isolated from cell conditioned media, plasma or urine[10,38]. While a low MWCO can result in co-isolation of soluble proteins, a high MWCO can trap small EVs leading to reduced recovery. Overall, UF-MWCO had limited impact on RNA yield and composition. As expected, as MWCO increased protein yield decreased. Importantly, the highest EV purity was observed with UF-100kDa based on the ratio particle to protein concentration[18]. The superior purity of UF-100kDa EV preparations was further confirmed by transmission electron microscopy. We also tested speed vacuum as an alternative method to concentrate EVs post-SEC which, to our knowledge, has not been investigated. Speed vacuum is commonly utilized after protein digestion to dry peptides for proteomic analysis with minimal loss. Particle recovery in samples concentrated with speed vacuum was similar to those concentrated with ultrafiltration. However, contaminants including aggregates were visible by transmission electron microscopy. The speed vacuum method resulted in the highest protein yield, ~3 times higher than UF-100kDa.

However, mass spectrometry revealed that speed vacuum samples had decreased protein diversity ( ~ 45% less proteins detected) compared to samples prepared with ultrafiltration. This may be due to high concentrations of abundant soluble contaminants and protein degradation during the drying process. Based on these findings, we do not recommend the use of speed vacuum to concentrate stool-derived EVs following SEC. Altogether, these findings demonstrate that ultrafiltration MWCO affects purity, but has limited impact on particle recovery, protein composition, and protein/RNA yield. This is likely due to SEC efficiently removing soluble proteins from stool supernatant prior to EV concentration.

We also isolated stool-derived EVs from 63 individuals using a combination of SEC and UF-100kDa and characterized their protein composition by mass spectrometry. Following annotation with the human body proteome map, stool-derived EVs originate from the gastrointestinal tract with the large majority coming from the pancreas and at a lower extent from the small intestine, colon, liver, esophagus and stomach. Tissue annotation showed that spleen-derived proteins were over-represented in stool-derived EVs, and functional enrichment revealed these proteins were all associated with neutrophil activity. Interestingly, we identified several key components of neutrophil extracellular traps (NETs) including neutrophil elastase (ELANE), myeloblastin (PRTN3), myeloperoxidase (MPO) and lactotransferrin (LTF)[39]. NETs are sticky web-like structures composed of decondensed chromatin filaments released by neutrophils to trap and kill various bacteria and pathogens[40,41]. In stool-derived EV preparations, filaments and globular domains of NETs can be visualized by electron microscopy at high magnification. Neutrophils are critical regulators of the innate immune response in the gut and NETs have been previously found in feces of patients with auto-immune and infectious diseases[42,43]. The clinical investigation of stool-derived NETs as biomarker for CRC is warranted.

The protein composition of stool-derived EVs was very distinct from other biofluids such as plasma and urine. Albumin and apolipoproteins are commonly co-isolated proteins with blood-derived EVs, but they were undetectable in stool. Similarly, proteins involved in the coagulation and complement cascades are abundant in EVs prepared from blood and urine but significantly reduced in stool. In contrast, the proteome of stool-derived EVs is dominated by immunoglobulins and hydrolases, specifically pancreatic digestive enzymes. Immunoglobulins may play a critical role in the regulation of the immune system in the digestive tract. Specifically, immunoglobulins IgA are abundantly secreted at the mucosal surfaces of the gut creating a layer of immune protection against fungal infections and

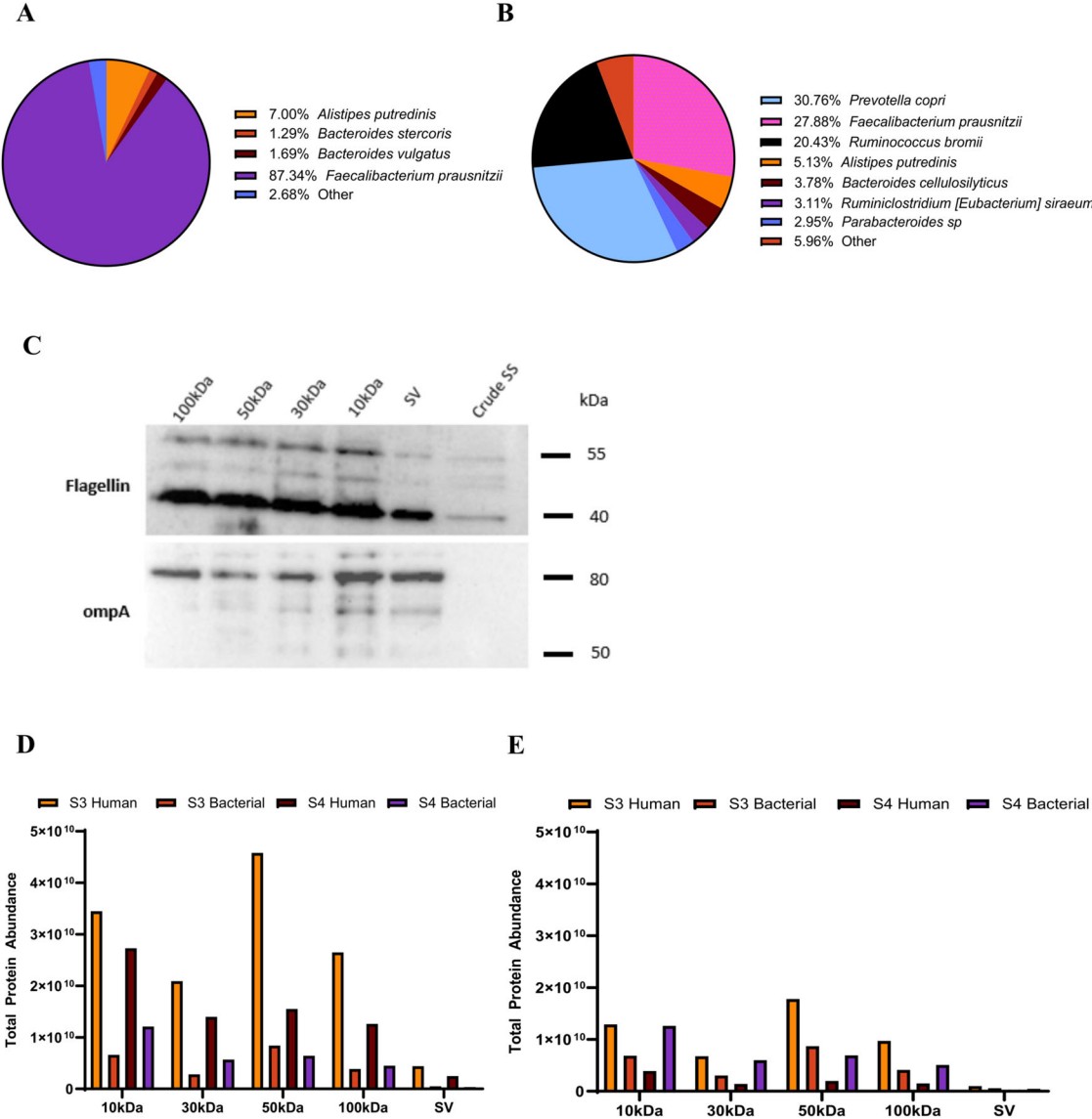

**Fig. 5 | Bacterial proteins identified in stool-derived extracellular vesicles. A** Most abundant species' in EV preparations from patient 3, UF 10 kDa sample. **B** Most abundant species in EV preparations from patient 4, UF 10 kDa sample. **C** Western blot (15ug) showing bacterial EV protein expression post-isolation for the five concentration methods and pre-isolation sample (stool supernatant). **D** Total human and bacterial protein abundances for each patient and concentration method. **E** Total human and bacterial protein abundances for each patient and concentration method after the top four human secreted proteins were removed from the calculation.

inflammation[44]. Two pancreatic proteases (CELA3A, CTRC) account for an average of 40% of the total protein abundance in stool EV preparations. Both proteins have a predicted molecular weight of 30 kDa; hence they are expected to be removed by ultrafiltration at a MWCO above 30 kDa. However, our data revealed the abundance of these proteins is higher in stool EVs compared to soluble fractions and unprocessed stool indicating that they are actually enriched during the EV isolation process. While these enzymes are annotated as secreted proteins, our dot blot data indicates that they are located within vesicles, specifically zymogen granules.

Digestive enzymes are packaged in zymogen granules by pancreatic acinar cells before being released into the pancreatic duct for transport to the duodenum[45]. The function of zymogen granules is to protect protease-producing cells from enzymatic degradation. Once secreted from the cell, the zymogen is cleaved, and their contents/active enzymes are released via exocytosis in a process similar to exosome biogenesis[46,47]. This is why we can find these enzymes within and outside vesicles. Several key proteins involved in zymogen granule formation were found in the stool EV proteome including the major zymogen granule membrane protein GP2, the protein scaffold ZG16, and the integral membrane-associated protein 1 (ITMAP1/CUZD1)[23–30]. The presence of intact GP2+ zymogen granules was further confirmed by dot blot and nanoscale flow cytometry. Besides the pancreas, ZG16 has been reported to be highly expressed in the normal colon but downregulated progressively during CRC development from precursor adenomatous polyps to adenocarcinoma[32,48,49]. The role of stool-derived ZG16 as diagnostic biomarker in CRC remains to be explored. Interestingly, basic salivary proline-rich proteins (PRB1, PRB3, PRB4, PRH1, SMR3B) were also detected in stool EV preparations. These enzymes are involved in food digestion and protection from oral pathogens. Similar to pancreatic enzymes, proline-rich proteins are released in zymogen granules by salivary glands[50]. The zymogen granule 16 homolog B (ZG16B), expressed in salivary zymogen granules was also detected in the stool EV proteome[51,52]. Altogether, our data reveal for the first time the co-isolation of stool-derived EVs with zymogen granules originating from the pancreas and salivary glands. Given that pancreas-derived components are found abundantly in

stool, it may serve as a promising noninvasive biomarker source for the detection of pancreatic cancer.

Several colon-enriched proteins were identified in stool, although at lower abundances than proteins originating from the pancreas. Notably, we identified 10 proteins with subcellular location at the plasma membrane including three proteins (SLC26A3, LGALS4 and CEACAM5) with specificity to the colon. These surface markers can be potentially used for the enrichment of colon-specific EVs using immunocapture strategies[53]. SLC26A3/DRA is an anion transporter expressed predominantly in the mucosa of the lower gastrointestinal tract and is downregulated in polyps, colon tumors, and inflammatory bowel disease[54–56]. Galectin-4 (LGALS4) is mainly expressed in the luminal epithelia of the gastrointestinal tract and is reported to have tumor suppressive effects. At both the mRNA and protein level, LGALS4 is significantly downregulated in colorectal adenomas and was nearly undetectable in invasive carcinomas[57–59]. Zhang and others (2023) detected galectin-4 in fecal EV samples from both CRC and normal patient groups[15]. CEACAM5 is a cell surface protein that is upregulated in colorectal cancer and has been functionally associated with tumor differentiation, invasion, and metastasis[60,61]. Antibodies targeting CEACAM5 are currently investigated for molecular imaging and T-cell directed therapies in human CRC[62–65]. In our study, we successfully detected CEACAM5-positive EVs by nanoscale flow cytometry revealing promising opportunities for novel non-invasive diagnostic tools in CRC.

EVs are released by all three domains of life (eukaryotes, bacteria, and archaea) and represent a universal, evolutionarily conserved mechanism[12]. Specifically, bacterial biomass is the main component (25–54% of dry solids) of the organic phase in feces[11]. The bacterial composition of stool is complex and dynamic and varies greatly among individuals. In the current study, there was a vast diversity of bacterial proteins from multiple species, but surprisingly human-derived proteins made up a larger portion of the total protein abundance. The microbiome proteomic landscape in stool-derived EVs has been examined previously in healthy, Crohns', and a variety of cancer patients[13,14]. For the concentration method comparison samples, we identified many bacterial proteins from the top previously recognized phyla in the gut: Firmicutes and Bacteroidetes[66]. The top bacteria species in these phyla included: *Faecalibacterium prausnitzii, Bacteroides vulgatus, Prevotella copri, Parabacteroides*, and *Alistipes putredinis*. Additionally, we determined highly expressed bacterial proteins (OmpA family protein, TonB-linked outer membrane protein, and SusD family protein) that were conserved across many of the bacteria found in stool EV preparations. These may be strong marker candidates for the separation of bacteria-derived EVs from host-derived EVs. Further exploration is necessary in order to gain novel biological insights on microbiota-host interactions and the impact on pathophysiological processes such as colon cancer.

We recognize there are some limitations associated with this study. For the concentration method comparison dataset, the top eight most abundant proteins within each sample covered 66–96% of the total protein mass for EV preparations. These highly abundant human-derived proteins may limit the ability to detect minimally expressed biologically relevant proteins. Additionally, the comparative study of ultrafiltration MWCO was conducted on only two patient samples which precludes robust statistical analysis and strong conclusions. However, little to no variation was observed between MWCOs tested, therefore we do not anticipate a different outcome with a larger sample size. Transcriptomic profiling of CRC has led to the creation of a consensus molecular subtype (CMS) classification system which can improve prognostication and guide treatment selection[67,68]. Our study did not include a transcriptomic analysis of stool-derived EVs, but we acknowledge the relevance of characterizing the RNA content of stool-derived EVs and determining the potential of a stool-based CMS classification. Given the tissue heterogeneity and diversity of stool-derived EVs, our proteomic data support the need to develop analytical tools to interrogate the content of colon-derived EVs with high sensitivity and specificity.

In summary, our proteomics studies provide several important findings that will serve as framework for future stool EV biomarker discovery in CRC and other GI-tract malignancies. First, we validated the combination of size-exclusion chromatography and ultrafiltration to isolate stool-derived EVs with acceptable recovery and purity. Second, we identified the tissue of origin of stool-derived EVs and their relative contribution to the total EV population. Notably, we revealed the co-isolation of pancreatic and salivary zymogen granules as well as neutrophil extracellular traps in stool EV preparations. Finally, we identified several surface markers of colon-derived EVs that could potentially serve as targets for isolation of tissue-specific EVs.

## Materials and methods

The detailed methodology used for EV isolation and characterization has been added to EV-TRACK (ID: 240195).

### Human specimens

This work was approved by the Mayo Clinic Institutional Review Board with informed patient consent under IRB 18-008752 and all ethical regulations relevant to human research participants were followed. For concentration method comparisons, stool specimens from cancer-free individuals ($n = 4$; 3 female, 1 male; age: $58.27 \pm 5.02$) were obtained from an existing prospectively enrolled Institutional Review Board approved archive at Mayo Clinic Rochester Minnesota. These stool specimens were collected between 2019 and 2020. The stool specimens from the larger cohort of patients used for proteomics were collected in 2018-2023 ($n = 63$, Table 2). Patients had no dietary restrictions, and no information on antibiotic usage and last meal was collected. Stool was collected from cancer patients, prior to them undergoing treatment. Patients collected stool specimens in sample container, and tris-EDTA based preservative buffer was added to stool in a 1:5 ratio, aliquoted into multiple 50 ml volume conical tubes, and stored in a − 80 °C freezer within 3 days. Samples were thawed overnight at 4 °C. They were mixed by vortexing for 30 s, and then centrifuged at $4500 \times g$ for 45 min. The supernatant was then transferred to a 50 ml conical tube. Stool supernatant (14 ml) was transferred to a 15 ml conical tube (Fischer Scientific, Waltham, MA, USA), and one polyvinylpolypyrrolidone (PVPP) tablet was added. The samples were then vortexed for 1 min and placed on a platform rocker for 15 min at room temperature. The mixture was transferred to a filter (8247 Frit) and centrifuged at $3300 \times g$ for 6 min. The filtered and treated stool supernatant was stored at −80 °C until EV isolation.

Pancreatic juice samples served as positive controls for pancreatic enzyme detection in stool. Pancreatic juice samples ($n = 6$; 2 cancer-free, 1 stage III pancreatic ductal adenocarcinoma, 1 chronic pancreatitis, 2 intraductal papillary mucinous neoplasms) were collected from patients undergoing a clinically indicated endoscopic ultrasound (EUS), endoscopic retrograde cholangiopancreatography (ERCP) or upper gastrointestinal endoscopy (EGD) procedure. Endoscopic procedures were performed after an overnight fast and under conscious sedation or anesthesia per institutional clinical standard of care. Gastric fluid was aspirated and discarded before intubation of the pylorus to minimize contamination. Synthetic human secretin (ChiRhoClin Inc, Burtonsville, MD) at a dose of 0.2 µg/kg (or, if the subject weighs more than 80 kg, a total dose of 16 µg) was administered intravenously over 1 min while the endoscope is positioned in the 2nd portion of the duodenum. From within the duodenum and without cannulation of the papilla of Vater, a 2.3-mm plastic aspiration catheter (Olympus, Tokyo, Japan) was passed through the biopsy channel of the endoscope until visible on screen in the endoscopic monitor. Fluid was continuously aspirated from the duodenum starting at the conclusion of secretin administration and continuing for 10 min. Buffer solution was added to the pancreatic juice immediately following collection and they were stored at −80 °C. Samples were then thawed, centrifuged at $2000 \times g$ for 10 min at 4 °C, and the supernatant was collected. This work was approved by the Mayo Clinic Institutional Review Board with informed patient consent under IRB 18-011965.

### EV isolation

Stool supernatant was centrifuged at $13,000 \times g$ for 3 min prior to isolation to remove any cellular debris that was not removed during initial processing[5]. Extracellular vesicles were isolated from stool supernatant using size-

**Table 2 | Clinical characteristics for patients in the larger cohort mass spectrometry experiment**

| Characteristics | Cancer Patients (n = 16) | Precancerous Lesion Patients (n = 32) | Cancer-Free Patients (n = 15) |
|---|---|---|---|
| Age [median (range)] | 57 (38–79) | 65 (51–77) | 63 (42–76) |
| Sex (Male/Female) | 11/5 | 20/12 | 4/11 |
| Smoking Status* | | | |
| Never | 9 | 16 | 11 |
| Current | 0 | 4 | 1 |
| Past | 7 | 12 | 3 |
| Alcohol Use | | | |
| Yes, past use, but not currently | 5 | 4 | |
| Yes, currently less than 3 times per week | 10 | 15 | 9 |
| Yes, currently 3 or more times per week | 0 | 6 | 4 |
| No | 1 | 5 | 2 |
| Unknown | | 2 | |
| Site | | | |
| Left-sided | 14 | | |
| Right Sided | 2 | | |
| Unknown | 0 | | |
| Lesion Classification | | | |
| Tubular Adenoma | | 9 | |
| Tubulovillous Adenoma | | 7 | |
| Sessile Serrated Lesion | | 16 | |
| Size (cm; range: 0.7–8.1)** | | | |
| ≤2.75 | 7 | 26 | |
| >2.75 | 7 | 6 | |
| Unknown | 2 | 1 | |
| Stage | | | |
| I | | 5 | |
| II | | 2 | |
| III | | 6 | |
| IV | | 3 | |

*Current: currently or quit in last 3 months, Past: past use, but not in last 3 months.
**If multiple tumors/lesions present, median size was used.

exclusion chromatography [(SEC; qEV1/35 nm & 70 nm), IZON Science (Christchurch NZ)] following the manufacturer's instructions. The columns were equilibrated with 13.5 ml of PBS prior to using. Then 1 ml of stool supernatant was pipetted onto the column and collected into a tube. Each subsequent fraction (700 μl) was collected in a separate tube. For the concentration method comparison, protein rich fractions (F18-F20) were pooled for mass spectrometry. Nanoscale flow cytometry (nFCM) or microfluidic resistive pulse sensing (MRPS) was performed on pooled SEC EV fractions to determine recovery and particle size distribution (PSD).

## EV concentration
SEC fractions (F7-F11; 3.5 ml) were further concentrated by either ultra-filtration (UF) or speed vacuum (SV). Amicon ultra-4 ml and 15 ml regenerated cellulose centrifugal filter devices (Millipore Sigma, Amicon Ultra device, Burlington MA) with molecular weight cutoffs of 10 kDa, 30 kDa, 50 kDa, or 100 kDa were assessed. For UF, samples were centrifuged

at $4000 \times g$ per the manufacturer's recommendation until the concentrate volume was around 200 μl. A pipette was used to aspirate the concentrate in the filter device, and the volume was recorded. For speed vacuum concentration, samples were centrifuged in a Savant DNA 130 SpeedVac (Thermo Scientific, Waltham MA) at 35 °C to avoid any heat degradation, until the volume was around 200 μl. MRPS was performed on concentrates to determine post-SEC/UF recovery and total recovery. For the larger cohort, all stool EV samples were further concentrated with UF-100kDa filters.

## Transmission electron microscopy
Transmission electron microscopy was carried out following our previous protocol at Mayo Clinic's Microscopy and Cell Analysis Core[5]. Extracellular vesicles were fixed in an equal volume of 4% paraformaldehyde /0.1 M phosphate buffer overnight at 4 °C. Sample (5 μl) was placed on a formvar-carbon-coated nickel grid (200 mesh), air dried for 30 min, and washed with PBS (6 × 3 min). The samples were then fixed in 1% glutaraldehyde/PBS for 5 min and washed with distilled water (6 × 3 min). They were transferred to a mixture of 4% uranyl acetate (freshly made) and 2% methylcellulose (in 1:9 ratio) for 5 min. Filter paper was used to soak up the solution that was left on the formvar coated grids and they were air dried for 1 h. Samples were observed with a JEOL (Peabody MA) 1400 electron microscope.

## Nanoscale Flow Cytometry (nFCM)
Pooled SEC EV fractions from 75 nm and 35 nm columns and ultrafiltration concentrates from 4 ml and 15 ml centrifugal filters were analyzed by an A60-MicroPlus nanoscale flow cytometer (Apogee Flow Systems Inc., Hertfordshire UK). Before sample analysis, the A60-MicroPlus was calibrated using a Rosetta calibration bead mix (Exometry Inc., Amsterdam, Netherlands) as previously described[69]. Side scatter triggering threshold was set at 1700 a.u corresponding to a scattering cross-section of 12 nm$^2$ and a particle diameter of 158 nm (Refractive Index core = 1.38; Refractive Index shell = 1.48; shell thickness = 4 nanometers). To determine percentage recovery, samples were diluted in sterile PBS and were ran at a flow rate of 1.5 μl/min for 1 min with an event rate below 7000 events per second to avoid swarm effect. Before each run, nFCM underwent a quality control procedure including a run with a mix of polystyrene and silica polydisperse beads (Apogee bead mix #1493, Apogee Flow Systems) to control for instrument sensitivity and flow rate stability. Buffer-only control (sterile PBS) was analyzed with the same instrument/acquisition settings and the event rate was kept below 80 events per second. Additional information can be found in the MIFlow-Cyt document (Supplementary Table 1).

Ten microliters of stool supernatant and pancreatic juice were pre-diluted 1/40 and 1/10 respectively. Samples were incubated with fluorescent antibodies (final concentration: 0.01 mg/mL) against CEACAM5 (NBP2-54625G DyLight 488, Novus Biologicals) and/or GP2 (D277-3, MBL International) for 30 min at room temperature in the dark. GP2 was conjugated to Alexa Fluor™ 647 using a previously published protocol[70]. DPBS was added to stop labeling reaction, and samples were analyzed immediately on an A60-MicroPlus nanoscale flow cytometer (Apogee Flow Systems Inc., Hertfordshire UK). Fluorescent antibodies in buffer (DPBS) were used as a negative control.

## Microfluidic Resistive Pulse Sensing (MRPS)
nCS1 (Spectradyne, Signal Hill, CA) was used to conduct MRPS measurements of particle size distribution (PSD) from crude stool supernatant, pooled SEC fraction, and concentrate samples. A solution of 1% Tween 20 (v/v) in DPBS was prepared to both prime the instrument and dilute each sample for analysis. Samples were initially diluted (1:100-1:1000) in DPBS, and then underwent a second dilution (1:2) in DPBS + 0.1% Poloxamer 188 prior to analysis. The instrument parameters (e.g. pressure at each port of cartridges) were determined as described previously[71]. We loaded 5μl of each sample onto polydimethylsiloxane cartridges. We used both C-400 (65 to 400 nm) and C-900 cartridges (130 to 900 nm) to determine the PSD of particles in each sample.

For post-acquisition analysis we used nCS1 Data Viewer (Version 2.5.0.319, Spectradyne). The peak filters were applied to all single particles individually to exclude false positive signals. All raw data were coalesced after processing by applying additional filters and subtracting background noise, as recommended by the manufacturer, and suggested previously[71]. After manual data analysis, PSDs were exported to excel files with 10-nm bins.

## RNA composition

For each concentration method, the concentrate was divided into two equal volume aliquots. Total RNA was extracted from one aliquot using the miRNeasy Mini Kit (Qiagen, Germantown MD) following the manufacturer's instructions. The on-column DNase digestion with the RNase-Free DNase Set was performed. The other aliquot underwent proteinase K and RNase A treatment to remove all non-vesicular RNA prior to RNA extraction as described previously[72]. Each RNA sample (1 µl) was loaded on an Agilent Pico RNA chip (Agilent Technologies, Santa Clara CA) to determine RNA concentration and fragment size distribution.

## Protein Concentration Assay (BCA)

Samples were incubated with Pierce RIPA buffer and Halt protease inhibitor cocktail (Thermo Scientific, Waltham MA) at 4 °C for 30 min[5]. Samples were then centrifuged at 13,000 × $g$ for 5 min at 4 °C, and supernatant containing protein lysates were aspirated[5]. Protein concentrations were determined by Pierce BCA Protein Assay Kit (Thermo Scientific, Waltham MA) following the manufacturer's instructions.

## Sample preparation and mass spectrometry

There was a total of 76 samples submitted and processed according to Mayo Clinic's Proteomics Core's procedures[73,74]. For the concentration method comparison, there were two patients (patient 3 and 4) with six samples each which included: 10 kDa, 30 kDa, 50 kDa, 100 kDa, SV, a pooled soluble fraction from SEC (SF: F18-F20), and a crude stool supernatant sample (crude SS: patient 3 only). For the larger cohort, there was a total of 63 patients. Dried sample pellets were resuspended in 50 mM TEABC/5% SDS and then subjected to in-solution tryptic digestion on a S-Trap column (ProtiFi) per the manufacturer's instructions. The resulting peptides were then analyzed by LC-MS/MS on a high-resolution Orbitrap Exploris 480 mass spectrometer (Thermo Scientific, Waltham MA). Briefly, each digest of peptides was resuspended in 0.1% trifluoroacetic acid, 0.2% formic acid, and 0.002% zwittergent 3-16, loaded onto an Halo Es-c18 trap column (100 mm × 2 cm, Fisher Healthcare) at a flow rate of 8 µl/min and separated on a C18 column (2.2 µm, 75 µm × 30 cm) heated at 50 °C at a flowrate of 300 nl/min using a gradient of buffer (5% solvent B from 0 to 5 min, 5% to 45% B from 5 to 125 min, 45% to 95% B from 125 to 128 min, 95% B from 128 to 132 min, 95% to 5% B from 132 to 134 min, and held at 5% from 134 to 137 min; solvent A: 0.1% formic acid and 2.5% acetonitrile; solvent B: 0.1% formic acid, 80% acetonitrile, and 10% isopropyl alcohol) by an Ultimate 3000 system (Thermo Scientific, Waltham MA) coupled with a high-resolution Orbitrap Exploris 480 mass spectrometer (Thermo Scientific, Waltham MA). Ionization of eluting peptides was performed using a nano-flex source kept at an electric potential of 2.3 kV. All mass spectra were acquired in a data-dependent acquisition (DDA) mode with ion isolation window of 1.6 m/z, default charge state of +2 and only precursors with charge states ranging from +2 to +6 selected for MS/MS events. Fixed collision energy was set at 28%. MS precursor mass range was set to 340 to 1600 m/z. Automatic gain control and the max injection time for MS and MS/MS was set to the standard. A 25 sec dynamic exclusion was to avoid repetitive fragmentation of same species. Data acquisition was performed with an option of lock mass (445.1200025 m/z).

## Protein identification and bioinformatic analysis

For the concentration methods comparison, Peaks X Pro was used for de-novo analysis of peptide sequences[75]. These were mapped against RefSeq non-redundant protein sequence database[76]. A database was generated that

contained nearly 100 million protein sequences from 262 "razor" species (many of them related to food). These results were too large for a conventional database search, so we mapped sequences to gastrointestinal tract proteins from the Human Microbiome Reference Genome Database (457 species)[77]. There were 256720 protein sequences from 61 "razor" species detected. The database, enriched by SwissProt 2023_01 human protein sequences, was then used in MaxQuant database search at 1% FDR[78]. Proteins with less than two unique peptides detected were filtered out of the dataset. Additionally, for human data we performed tryptic and semi-tryptic peptide analyses. Peptides that ends with R or K in one end and any amino acid at the other end were defined as semi-tryptic peptides.

For the larger cohort of samples, similar methods were applied, but human-derived proteins were the primary focus. Proteins with less than two peptides detected were filtered out of the dataset. Human Protein Atlas was used to classify subcellular location. STRINGdb was used to evaluate the relationships among detected proteins and assess enriched functions and pathways. To further evaluate soluble protein contamination, proteins were annotated to the human secretome[79]. The human body proteome map was used to further characterize the tissue of origin of stool-derived EV proteins[32].

## Western blot

NuPAGE LDS sample buffer (4x) and NuPAGE reducing agent (10x) (Fischer Scientific, Waltham MA) were added to equal protein amounts (15ug) from each of the EV concentration methods and crude stool supernatant. Samples were run on NuPAGE 4–12%, Bis Tris, 1.0 m, mini protein gel (Fischer Scientific, Waltham MA) after boiling for 10 min at 70 °C. Proteins were transferred to nitrocellulose membranes using the iBlot 2 transfer device and were blocked with 5% BSA for 30 min. The membranes were then incubated overnight at 4 °C with the following antibodies: TSG101 (1:1000, Abcam, ab30871), CD63 (1:1000, Abcam, ab231975), LGALS3BP (1:1000, Abcam, ab217572), ompA (1:5000, Antibody Research, 111120), Flagellin (1:10000, Abcam, ab93713), GP2 (1:100, Novus Biologicals, NBP3-07327), CELA3A (1:2000, GeneTex, GTX104918), SLC26A3 (1:200, Santa Cruz Biotechnology, sc-376187), LGALS4 (1:250, Abcam, ab229347) and CEACAM5 (1:1000, Cell Signaling Technology, 2383S). After incubation with primary antibodies, the membranes were washed with 0.1% tween in tris-buffered saline (TBST) three times for 5 min each. Incubation with secondary antibody was performed at room temperature with either goat anti-mouse or goat anti-rabbit IgG secondary antibody, HRP conjugated for 1 h. Membranes were washed with 0.1% tween in tris-buffered saline (TBST) three times for 10 min each. Membranes were exposed to Supersignal West Pico Plus chemiluminescent substrates (Fischer Scientific, Waltham MA) in a 1:1 ratio and incubated for 5 min prior to imaging on a Chemidoc XRS+ system (Bio-Rad Laboratories, Hercules CA). If proteins overlapped in size, Restore Western Blot Stripping Buffer (Thermo Scientific, Waltham MA) was used to strip antibodies from the membranes prior to reblotting.

## Dot blot

Dot blotting of EV preparations was performed similar to what was described previously[80,81]. Various concentrations of isolated stool -derived EVs were absorbed onto nitrocellulose membranes at room temperature for 1 h. The membranes were blocked with 5% BSA in the absence or presence 0.1% (v/v) Tween-20 (TBST) at room temperature for 1 h. Membranes were incubated with anti-GP2 or anti-CELA3A in TBS or TBST blocking buffer at 4 °C overnight followed by HRP-conjugated secondary antibody incubation at room temperature for 1 h. Blots were imaged using a Chemidoc XRS+ system (Bio-Rad Laboratories, Hercules CA).

## Statistics and reproducibility

Student $t$-test (parametric) or Mann–Whitney test (non-parametric) was employed to compare two groups. One-way ANOVA (parametric) or Kruskal–Wallis (non-parametric) tests were used to compare three or more groups. Further information on what statistical test was used is

provided in each figure legend. The results were considered significant for *p*-values < 0.05. *p*-values were either specified in the figure or denoted as asterisk (s): \*P < 0.05, \*\*P < 0.01, \*\*\*P < 0.001, \*\*\*\*P < 0.0001. All data were analyzed and plotted in GraphPad Prism 9.5.1 (Carlsbad CA). To measure recovery efficiency, we compared particle numbers from crude stool supernatant (f0) and post-EV isolation (SEC recovery). Post-SEC/UF recovery was calculated by dividing particle number post-concentration and SEC total particle number. Lastly, total recovery was determined by dividing particle number post-concentration by particle count of crude stool supernatant (f0). Venn diagrams were created using https://bioinformatics.psb.ugent.be/webtools/Venn/.

## Reporting summary

Further information on research design is available in the Nature Portfolio Reporting Summary linked to this article.

## Data availability

The authors declare that most of the main data supporting the conclusions of the current study is available within the manuscript and its supplementary files. The numerical source data for graphs and charts is shown in Supplementary Data. The uncropped western blot and dot blot results are shown as Supplementary Figs. 12 and 13. This study includes no data deposited in external repositories. For access to other data contact the corresponding author. It may be made available, but it is conditional to approval from the Mayo Clinic Institutional Review Board and will require a data use agreement from Mayo Clinic Legal Contracts Administration.

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

## Acknowledgements

We would like to thank Mayo Clinic's Proteomics Core and Mayo Clinic's Microscopy and Analysis Core for their assistance on this project. This work was supported by the National Cancer Institute under CA214679 to JBK and Exact Sciences (Madison WI). The graphical abstract was created in BioRender: Northrop, E. (2025) https://BioRender.com/n22s427.

## Author contributions

Conceptualization: W.R.T., J.B.K., F.L., Data curation: E.J.N.A, Y.K., F.L., Formal analysis: E.J.N.A, F.L., Visualization: E.J.N.A., F.L., Methodology: E.J.N.A., F.L., Investigation: E.J.N.A, Y.K., Funding acquisition: J.B.K., Project administration: E.J.N.A., Resources: J.B.K., F.L., S.M., Writing—original draft: E.J.N.A., F.L., Writing—review and editing: all authors.

## Competing interests

The authors declare the following competing interests: W.R.T. and J.B.K. are inventors of Mayo Clinic intellectual property, licensed by Exact Sciences (Madison WI), from which royalties may be paid to Mayo Clinic. In addition to NIH grants CA214679 and CA241164, J.B.K. receives funding from a sponsored research agreement between Mayo Clinic and Exact Sciences. All other authors declare no competing interests.
