## [Transparent Peer Review file · Communications Biology]

The proteomic landscape of stool-derived extracellular vesicles in patients with pre-cancerous lesions and colorectal cancer

Corresponding Author: Dr Emmalee Northrop-Albrecht

Version 0:

Reviewer comments:

Reviewer #1

(Remarks to the Author)

The study is focused on EV isolation methods from stool of patients with colorectal cancer and controls individuals. They demonstrate that concentration methods post-SEC have limited impact on stool EV derived protein and RNA recovery. They also claim to identify potential markers of precancerous lesions by analyzing the bacterial and host proteome of stool EV preparations from cancer-free individuals and patients with different stages of CRC. Additionally, they found that zymogen granules containing highly abundant enzymes are likely co-isolated in stool EV preparations. Finally, they were able to quantitatively compare proteins from bacteria and human in stool samples.

This is interesting and novel work and compares different EV concentrations methods, which denotes rigor of the investigation and is important for the scientific community. However, the description of the results of the proteomic analysis would benefit from a more immediate and clear description of the differences between cancer and non-cancer individuals. Also, a clear table of the cohort would be helpful. Additionally, with the current data, it is unclear if the presence of pancreatic enzymes is due to the release of EVs or simply EV-free enzymes from the pancreas co-isolated with the EVs.

The discussion starts with biomarkers. However, this is not a biomarker manuscript but rather a study that focuses on the pre-analytical variables for isolation of stool-derived EVs. Also, the discussion focuses, for the most part, on repeating the result section rather than embedding them in the discussing the significance and implications.

In conclusion, this is an interesting study that could be streamlined highlighting the relevance of the findings in the context of colon cancer.

Reviewer #2

(Remarks to the Author)

General comments:

This study employs and compares different EV concentration methods from human stool samples. EV proteomes were also investigated. The novelty and strength of this study are in the thorough characterization of stool EVs using different robust methods. The comprehensive assessment of the stool-derived EV proteome in healthy controls as well as colorectal cancer patients showed the presence of colon-associated protein profiles that could pave the way for potential biomarker studies in the future. It is recommended that the methods of this study should be reported to the EV track (<http://evtrack.org>). The manuscript is publishable with some minor revisions as described below.

Specific Comments:

ABSTRACT

Line 44: Isolation of EVs from stool supernatant?

Line 58: The keyword Extracellular Vesicles/classification should be either two different keywords or written just as 'Extracellular Vesicles'

INTRODUCTION

1. Lines 80-81: The context of this statement isn't clear. Is stool referred to as a biofluid here?
2. Lines 93-95: The proteome landscape of stool EVs has been investigated in other studies as well which should be discussed here and state how your study is different or unique. Below are some of the references for your consideration-

I. Mishra S, Tejesvi MV, Hekkala J, Turunen J, Kandikanti N, Kaisanlahti A, Suokas M, Leppä S, Vihinen P, Kuitunen H, Sunela K, Koivunen J, Jukkola A, Kalashnikov I, Auvinen P, Kääriäinen OS, Peñate Medina T, Peñate Medina O, Saarnio J, Meriläinen S, Rautio T, Aro R, Häivälä R, Suojanen J, Laine M, Erawijattari PP, Lahti L, Karihtala P, Ruuska TS, Reunanen J. Gut microbiome-derived bacterial extracellular vesicles in patients with solid tumors. *J Adv Res.* 2024 Mar 7:S2090-1232(24)00090-0. doi: 10.1016/j.jare.2024.03.003. Epub ahead of print. PMID: 38458256.

II. Zhang Z, Liu X, Yang X, Jiang Y, Li A, Cong J, Li Y, Xie Q, Xu C, Liu D. Identification of faecal extracellular vesicles as novel biomarkers for the non-invasive diagnosis and prognosis of colorectal cancer. *J Extracell Vesicles.* 2023 Jan;12(1):e12300. doi: 10.1002/jev2.12300. PMID: 36604402; PMCID: PMC9816085.

III. Zhang, X., Deeke, S.A., Ning, Z. et al. Metaproteomics reveals associations between microbiome and intestinal extracellular vesicle proteins in pediatric inflammatory bowel disease. *Nat Commun* 9, 2873 (2018). <https://doi.org/10.1038/s41467-018-05357-4>

METHODS

3. Lines 117-118: Do you have any information on antibiotic usage by the study participants and cancer treatment in the case of cancer patients? If yes, that information should be provided.

4. Line 144: Please provide relevant references for the methodology

5. Line 154: Please provide relevant references for the methodology

6. Line 165: Please provide relevant references for the methodology

7. Line 219: Please provide relevant references for the methodology

8. Line 225: Please provide relevant references for the methodology

9. Line 236: Is this -4 to 5 min?

RESULTS

Line 365: Supplementary Figure 1: The scale bar's unit is written incorrectly. Instead of 'uM', it should be 'µm' and in place of 'nM', it should be 'nm'.

DISCUSSION

Line 527: Stool shouldn't be referred to as a biofluid. Instead, biospecimen.

Version 1:

Reviewer comments:

Reviewer #1

(Remarks to the Author)

The authors have addressed all the comments.

Reviewer #2

(Remarks to the Author)

The manuscript has been revised with the incorporation of suggested corrections and valid explanations. It is now apt to be processed for publication.

We thank the reviewers for their time and valuable suggestions to improve our manuscript. We have carefully considered the comments and have made the suggested changes to the manuscript. Critiques raised by each reviewer have been addressed point-by-point below.

Reviewer #1 (Remarks to the Author):

The study is focused on EV isolation methods from stool of patients with colorectal cancer and controls individuals. They demonstrate that concentration methods post-SEC have limited impact on stool EV derived protein and RNA recovery. They also claim to identify potential markers of precancerous lesions by analyzing the bacterial and host proteome of stool EV preparations from cancer-free individuals and patients with different stages of CRC. Additionally, they found that zymogen granules containing highly abundant enzymes are likely co-isolated in stool EV preparations. Finally, they were able to quantitatively compare proteins from bacteria and human in stool samples. This is interesting and novel work and compares different EV concentrations methods, which denotes rigor of the investigation and is important for the scientific community. In conclusion, this is an interesting study that could be streamlined highlighting the relevance of the findings in the context of colon cancer.

- 1.) **The description of the results of the proteomic analysis would benefit from a more immediate and clear description of the differences between cancer and non-cancer individuals. A clear table of the cohort would be helpful.**

We have added **Table 1** below to the human specimens' methods section (line 120). The initial scope of this manuscript is to provide an overview of the proteome composition of stool EVs. While we agree with the reviewer of the relevance to compare cancer and non-cancer individuals, it goes beyond the focus of this study. We are currently analyzing the proteomic data among different cancer stages, and this data will be submitted for peer-review in a subsequent manuscript.

Characteristics	Cancer Patients (n=16)	Precancerous Lesion Patients (n=32)	Cancer-Free Patients (n=15)
Age [median(range)]	57 (38-79)	65 (51-77)	63 (42-76)
Sex (Male/Female)	11/5	20/12	4/11
Smoking Status*			
Never	9	16	11
Current	0	4	1
Past	7	12	3
Alcohol Use			
Yes, past use, but not currently	5	4	
Yes, currently less than 3 times per week	10	15	9
Yes, currently 3 or more times per week	0	6	4
No	1	5	2
Unknown		2	
Site			
Left-sided	14		
Right Sided	2		
Unknown	0		
Lesion Classification			
Tubular Adenoma		9	
Tubulovillous Adenoma		7	
Sessile Serrated Lesion		16	
Size (cm; range: 0.7-8.1) **			
≤2.75	7	26	
>2.75	7	6	
Unknown	2	1	
Stage			
I	5		
II	2		
III	6		
IV	3		

*Current: currently or quit in last 3 months, Past: past use, but not in last 3 months

** If multiple lesions/tumors present, median size was used

- 2.) **Additionally, with the current data, it is unclear if the presence of pancreatic enzymes is due to the release of EVs or simply EV-free enzymes from the pancreas co-isolated with the EVs.**

Based on proteomic analysis of SEC fractions, it appears that the pancreatic enzymes are inside and outside vesicles. In soluble protein-rich SEC fractions, pancreatic enzymes were detected albeit at a lower abundance compared to EV-rich fractions (**Figure 3C**). Dot-blot shows positive signal of CELA3A in intact EVs and the signal increased with Tween-20 detergent supporting the presence of pancreatic enzymes within and outside EVs (**Figure 3E**). In the discussion (lines 564-572), we indicated that the abundance of these secreted enzymes is higher in stool EVs compared to soluble fractions and unprocessed stool. Given this information we believe that these enzymes are packaged in zymogen granules, and these granules are coisolated during EV isolation. We can find these pancreatic enzymes within and outside granules because once zymogen granules are secreted from the cell, the zymogen is cleaved, and their contents/active enzymes are released via exocytosis in a process similar to exosome biogenesis.

- 3.) **The discussion starts with biomarkers. However, this is not a biomarker manuscript but rather a study that focuses on the pre-analytical variables for isolation of stool-derived EVs.**

We have removed the first two sentences of the discussion on biomarkers. We agree that it is more appropriate to begin the discussion with emphasizing that there is limited research on the pre-analytical variables for isolation of stool-derived EVs and the overall proteomic landscape.

- 4.) **Also, the discussion focuses, for the most part, on repeating the result section rather than embedding them in the discussing the significance and implications.**

We agree, there are some parts of the results that are repeated in the discussion. We removed conclusion statements in the results section and instead included it in the discussion and expanded on the significance and implications of our findings.

Reviewer #2 (Remarks to the Author):

This study employs and compares different EV concentration methods from human stool samples. EV proteomes were also investigated. The novelty and strength of this study are in the thorough characterization of stool EVs using different robust methods. The comprehensive assessment of the stool-derived EV proteome in healthy controls as well as colorectal cancer patients showed the presence of colon-associated protein profiles that could pave the way for potential biomarker studies in the future. The manuscript is publishable with some minor revisions as described below.

- 1.) **It is recommended that the methods of this study should be reported to the EV track (<http://evtrack.org>).**

We have uploaded our methods to EV-TRACK (ID: 240195). We also added the following statement in the methods section (line 111):” *The detailed methodology used for EV isolation and characterization has been added to EV-TRACK (ID: 240195).*”

- 2.) **ABSTRACT: Line 44: Isolation of EVs from stool supernatant?**

We have added “EV isolation method...” to line 44.

- 3.) **ABSTRACT: Line 58: The keyword Extracellular Vesicles/classification should be either two different keywords or written just as 'Extracellular Vesicles'**
Classification was removed from the keywords list.
- 4.) **INTRODUCTION: Lines 80-81: The context of this statement isn't clear. Is stool referred to as a biofluid here?**
As you suggested below, we used "biospecimen" when referring to stool (line 80).
- 5.) **INTRODUCTION: Lines 93-95: The proteome landscape of stool EVs has been investigated in other studies as well which should be discussed here and state how your study is different or unique.**

Below are some of the references for your consideration-

I. Mishra S, Tejesvi MV, Hekkala J, Turunen J, Kandikanti N, Kaisanlahti A, Suokas M, Leppä S, Vihinen P, Kuitunen H, Sunela K, Koivunen J, Jukkola A, Kalashnikov I, Auvinen P, Kääriäinen OS, Peñate Medina T, Peñate Medina O, Saarnio J, Meriläinen S, Rautio T, Aro R, Häivälä R, Suojanen J, Laine M, Erawijattari PP, Lahti L, Karihtala P, Ruuska TS, Reunanen J. Gut microbiome-derived bacterial extracellular vesicles in patients with solid tumors. *J Adv Res.* 2024 Mar 7:S2090-1232(24)00090-0. doi: 10.1016/j.jare.2024.03.003. Epub ahead of print. PMID: 38458256.

II. Zhang Z, Liu X, Yang X, Jiang Y, Li A, Cong J, Li Y, Xie Q, Xu C, Liu D. Identification of faecal extracellular vesicles as novel biomarkers for the non-invasive diagnosis and prognosis of colorectal cancer. *J Extracell Vesicles.* 2023 Jan;12(1):e12300. doi: 10.1002/jev2.12300. PMID: 36604402; PMCID: PMC9816085.

III. Zhang, X., Deeke, S.A., Ning, Z. et al. Metaproteomics reveals associations between microbiome and intestinal extracellular vesicle proteins in pediatric inflammatory bowel disease. *Nat Commun* 9, 2873 (2018). <https://doi.org/10.1038/s41467-018-05357-4>

We have added two of the papers to the introduction suggested by the reviewer (lines 94-102). Of the studies that have been conducted in this area of research, they differ in the type of disease studied, EV isolation method and protein detection assay used. Zhang and others (2023) used ultracentrifugation, which we found previously to be the most inferior isolation technique (Northrop-Albrecht et al., 2022). Additionally, they did not perform a discovery study but relied on previous literature for biomarker selection candidates. Mishra and others (2024) primary focus was on fecal bacterial EVs among several cancers, none of which were colorectal. And again, they used different EV isolation methods. Zhang and others (2018) performed metaproteomics on mucosal-luminal interface aspirates, not stool samples so we did not end up including it in the introduction. Overall, we tried to emphasize that stool EVs have been underexplored, and of the few studies that looked into the topic, none have performed a comprehensive analysis in relation to colorectal cancer and precancerous conditions like in our current study.

- 6.) **METHODS: Lines 117-118: Do you have any information on antibiotic usage by the study participants and cancer treatment in the case of cancer patients? If yes, that information should be provided.**
No information on antibiotic usage was collected from the patients in this study. Additionally, in the case of cancer patients, their stools were collected prior to any treatment therapy. This was added to the methods section (lines 120,121).
- 7.) **METHODS: Line 144: Please provide relevant references for the methodology**

We have referenced our previous methods paper that includes the additional centrifugation step that removes debris prior to EV isolation (Northrop-Albrecht et al., 2022). Otherwise, we followed the manufacturer's (Izon) instructions.

8.) METHODS: Line 154: Please provide relevant references for the methodology

We have added that we followed the manufacturer's instructions for the ultrafiltration process. Since EV concentration by speed vacuum has not been previously done, we decided to not run at a high temperature to avoid potential degradation.

9.) METHODS: Line 165: Please provide relevant references for the methodology

We added the reference from our previous methodology publication (Northrop-Albrecht et al., 2022).

10.) METHODS: Line 219: Please provide relevant references for the methodology

We added the reference from our previous methodology publication (Northrop-Albrecht et al., 2022).

11.) METHODS: Line 225: Please provide relevant references for the methodology

We have added two references to the sample preparation and mass spectrometry methods paragraph (line 230).

12.) METHODS: Line 236: Is this -4 to 5 min?

The text was changed to "5% solvent B from 0 to 5 min" in the manuscript (line 239).

13.) RESULTS: Line 365: Supplementary Figure 1: The scale bar's unit is written incorrectly. Instead of 'uM', it should be 'µm' and in place of 'nM', it should be 'nm'.

We changed the scale bar unit for **Figure 1** and **Supplementary Figure 1**.

14.) DISCUSSION: Line 527: Stool shouldn't be referred to as a biofluid. Instead, biospecimen.

We went through the manuscript and made sure we used the appropriate term.